# Anti-Malarial and Anti-Lipid Peroxidation Activities of Deferiprone-Resveratrol Hybrid in *Plasmodium berghei*-Infected Mice

**DOI:** 10.3390/biology10090911

**Published:** 2021-09-14

**Authors:** Hataichanok Chuljerm, Supawadee Maneekesorn, Voravuth Somsak, Yongmin Ma, Somdet Srichairatanakool, Pimpisid Koonyosying

**Affiliations:** 1Department of Medical Technology, School of Allied Health Sciences, Walailak University, Nakornsrithammarat 80160, Thailand; hataichanok.ch@wu.ac.th (H.C.); voravuth.so@wu.ac.th (V.S.); 2Research Excellence Center for Innovation and Health Products, Walailak University, Nakornsrithammarat 80160, Thailand; 3Department of Biochemistry, Faculty of Medicine, Chiang Mai University, Chiang Mai 50200, Thailand; maneekesorn@gmail.com (S.M.); somdet.s@cmu.ac.th (S.S.); 4School of Pharmaceutical and Chemical Engineering, Taizhou University, Taizhou 318000, China; yongmin.ma@tzc.edu.cn

**Keywords:** iron chelator, lipid peroxidation, malaria, *Plasmodium*, antioxidant

## Abstract

**Simple Summary:**

Malaria remains a public health problem in tropical and subtropical countries. The emergence of malaria parasite resistance to antimalarial drugs has been recently considered a serious issue. Alternative compounds have become an important therapeutic strategy to achieve malaria treatment. Iron chelators are widely used for the treatment of iron overload patients. The iron chelators also reveal an inhibitory effect on malaria parasite growth by depriving the parasite intracellular iron. This study presented the potential of the novel hybrid iron chelator, deferiprone-resveratrol hybrid on the inhibition of malaria parasite growth, the improvement of hematological parameters and the alleviatation of oxidative tissue damage in malaria-infected mice. Deferiprone-resveratrol hybrid would be used as a therapeutic/preventive compound to increase the efficacy of treatment and eliminate an antimalarial drug resistance.

**Abstract:**

Iron is essential for all organisms including fast-dividing malarial parasites. Inversely, iron chelators can inhibit parasite growth through the inhibition of DNA synthesis and can ameliorate oxidative cell damage. Deferiprone (DFP)-resveratrol (RVT) hybrid (DFP-RVT) is a lipophilic anti-oxidative, iron-chelating agent that has displayed potent neuroprotective and anti-plasmodium activities in vitro. The goal of this work was to investigate the inhibitory effects of DFP-RVT on parasite growth and oxidative stress levels during malaria infections. Mice were intraperitoneally infected with *P. berghei* and orally administered with DFP, DFP-RVT and pyrimethamine for 4 d. The percentage of parasitemia was determined using Giemsa’s staining/microscopic examination. Amounts of the lipid-peroxidation product, thiobarbituric acid-reactive substance (TBARS), were determined in both plasma and liver tissue. In our findings, DFP-RVT exhibited a greater potent inhibitory effect and revealed an improvement in anemia and liver damage in infected mice than DFP. To this point, the anti-malarial activity was found to be associated with anti-RBC hemolysis and the liver weight index. In addition, plasma and liver TBARS levels in the DFP-RVT-treated mice were lower than those in DFP-treated mice. Thus, DFP-RVT could exert anti-plasmodium, anti-hemolysis and anti-lipid peroxidation activities to a better degree than DFP in *P. berghei*-infected mice.

## 1. Introduction

Malaria is one of the three most deadly infectious diseases in the world. It is known to be caused by *Plasmodium* spp. parasites and transmitted from female *Anopheles* mosquitoes [1]. Several antimalarial drugs, such as pyrimethamine (PYR), chloroquine (CQ), artemisinin (ART), amodiaquine and mefloquine (MQ), have been widely used for the treatment of human malaria infections. However, the emerging resistance of the existing drugs has been reported and has become a major public health concern [2,3]. Liver dysfunction is recognized as a clinical manifestation of malaria infection in patients and effective anti-malarial drugs are required to improve liver function [4]. In terms of prophylaxis, some anti-malarial drugs, such as MQ, PYR and sulfadoxine, have been recommended for healthy people travelling to malaria endemic areas. However, these drugs are known to be the cause of reversible elevations in liver enzyme activity [5]. However, oral PYR treatment (10 mg/kg) has been found to restore plasma liver enzymes in *P. yoelii*-infected mice (5–10%) to almost normal levels [6,7].

Iron is a trace element that plays an important role in the cellular metabolism and proliferation of all living organisms, including *Plasmodium* parasites [8]. It is possible that malarial parasites obtain iron for growth and development from their hosts in the form of nonheme iron that is present in red blood cells (RBC), as well as from transferrin- and plasmodium siderophore-bound iron that is present in plasma [9,10,11,12]. Inversely, iron chelators withhold available iron from malarial parasites and subsequently inhibit parasite growth; therefore, they could potentially be candidates for adjuctive anti-malarial drugs in the treatment of malaria infections. Importantly, many iron chelators, such as deferoxamine (DFO), deferiprone (DFP), deferasirox (DFX), alkylthiocarbamates, 8-hydroxyquinoline, synthetic ferrichromes, FR160 catechol, 1-(N-acetyl-6-aminohexyl)-3-hydroxy-2-methylpyridin-4-one (CM1) and deferiprone-resveratrol (DFP-RVT) hybrid, have been administered in the treatment of *P. falciparum*-infected RBC (PRBC), while *P. bergheir*- and *P. vinckei*-infected mice have exhibited effective anti-plasmodium agents [10,13,14,15,16,17,18,19].

Actually, reactive oxygen species (ROS) can be beneficial for the host by defending against parasitic growth without causing oxidative cell damage. The amounts of short-chain lipid-peroxidation products, including alkanals, 4-hydroxyalk-2-enals, alka-2,4-dienals, 4-hydroxynon-2-enal, deca-2,4-dienal and hexanal, were found to increase in *P. vinckei*-infected RBC, while some of them were also found to be toxic to *P. falciparum* culture [20]. ROS is known to cause oxidative stress, which can lead to tissue damage and organ dysfunction in hosts [21,22]. Nonetheless, over-production of reactive oxygen species (ROS) and intraerythrocytic digested hemoglobin (hemozoin) during malaria infections results in decreased antioxidant protection and increased oxidant-mediated damage to tissues. Thus, the need to protect against oxidative stress in host organisms has become a concerning issue. A compound with broad therapeutic potential, such as an iron chelator and/or ROS scavenger, would be beneficial for the effective treatment of malaria. Lytton and colleagues previously synthesized lipophilic ferrichrome derivertives that exhibited a combination of fluorescent, anti-plasmodium and iron-chelating properties [12].

Herein, the multitarget directed ligand, DFP-RVT hybrid, has been synthesized through a merging of DFP with resveratrol (RVT) by Xu, et al. DFP-RVT is recognized by the chemical name 2-(3,5-dihydroxystyryl)-5-hydroxypyridin-4(1H)-one, the chemical formula C_13_H_11_NO_4_ and has a molecular weight of 245.23 g/mol. It possesses one iron-chelating component of DFP and another antioxidative component of RVT (Figure 1).

Interestingly, this compound has displayed better inhibition activities against Cu^2+^/Fe^3+^-induced Aβ_1–42_ aggregation than RVT and curcumin. It has also been found to exhibit more potent antioxidant activity than Trolox, but less potent antioxidant activity when compared with RVT. However, it did exhibit stronger metal-chelating activity than DFP [23]. Currently, we have demonstrated that the DFP-RVT hybrid was more efficient in decreasing LIP content in PRBC than DFP and was more effective in inhibiting parasitic growth than DFP and PYR in both mono- and combined-treatments [16]. This study focused on investigating the inhibitory effects of the DFP-RVT hybrid on parasitic growth and oxidative stress levels in *P. berghei*-infected mice.

## 2. Materials and Methods

### 2.1. Chemicals and Reagents

1-Butanol, butylated hydroxytoluene (BHT), DMSO, PYR, phosphate-buffered saline (PBS), 1,1,3,3-tetramethoxypropane (TMP), trichloroacetic acid (TCA), thiobarbituric acid (TBA) and Wright-Giemsa staining solution were purchased from Sigma-Aldrich Chemicals, Company (St. Louis, MO, USA). Additionally, the 1,2-Dimethyl-3-hydroxypyridin-4-one (DFP) was kindly provided by the Institute of Research and Development, Government Pharmaceutical Organization, Bangkok, Thailand.

### 2.2. Methods

#### 2.2.1. Deferiprone-Resveratrol Hybrid (DFP-RVT) Synthesis

DFP-RVT hybrid was synthesized by Professor Dr. Yongmin Ma, PhD. and colleagues from Zhejiang Chinese Medical University, Hangzhou, China. The design, synthesis processes and biological evaluation, with regard to its potent iron-binding and antioxidant properties, were established by Xu et al. [23].

#### 2.2.2. Animal Ethics

The protocol for animal experimentation in this study was approved of by the Ethics Committee for Animal Study, Walailak University, Tha Sala District, Thailand (WU-AICUC-64013).

#### 2.2.3. Animal Care

Male ICR mice (BW approximately 30 g) were purchased from Nomura Siam International Company (Limited), Bangkok, Thailand. The mice were housed separately with free access to food and water ad libitum under controlled temperatures (20–22 °C), humidity levels (50 ± 10%) and light (12 h light/dark cycle). The mice were acclimatized for one week before the experiment.

#### 2.2.4. Murine *P. berghei* Infection

*P. berghei* strain ANKA was obtained from the Malaria Research and Reference Reagent Resource Center (MR4), National Institute of Allergy and Infectious Diseases, Manassa, VA, USA. A cryo-frozen stock of PbANKA-infected RBC (PRBC) was thawed by being placed in a 37 °C water bath for 2–3 min, then diluted with normal saline solution to achieve PRBC suspension (10^7^ cellls/mL with a range of 5–20% parasitemia). It was then intraperitoneally (*ip*) injected into ICRmice. Tail vein blood samples were collected daily, prepared as thin-film smears on glass slides and stained with Wright-Giemsa staining solution. The numbers of PRBC were then counted up to a total of 500 cells under a light microscope (oil-immersion, 100× magnification). The mice were euthanized according to the institutional guidelines and blood was harvested via a cardiac puncture using a 25 guage ⅝′′ needle and a 1 mL plastic syringe. Blood was then collected in a lithium heparin tube and diluted in PBS solution to achieve PRBC suspension (1 × 10^7^/mL with 5–10% parasitemia) for passage of the infection and further experiments [24,25]. The percentage of parasitemia was calculated using the following formula:Parasitemia (%) = (Number of PRBC/Total numbers of RBC) × 100

#### 2.2.5. Suppressive Activity Test of DFP-RVT in *P. berghei*-Infected Mice

Stock solutions of PYR, DFP and DFP-RVT were freshly prepared in 100% DMSO. Dosages of the drugs (mg/kg BW) were adjusted according to the BW of mice using DMSO at a final concentration of 60%. Blood samples obtained from PbANKA-infected mice were diluted with PBS to reach PRBC suspension (1 × 10^7^ cells/mL with a range of 5–20% parasitemia) and were *ip* injected into experimental mice [26]. After that, the mice (n = 3) were orally administered with 60% DMSO, PYR, DFP (50 mg/kg BW) and DFP-RVT (50 mg/kg BW) on days 0, 1, 2 and 3 according to Peter’s 4 day test, as has been previously described by Knight and Peters with slight modifications [25,27]. The percentage of suppression of parasitic growth was calculated using the following formula [28]:Percentage of suppression = (%parasitemia_(untreated group)_−%parasitemia_(treated group)_) × 100% parasitemia_(untreated group)_

On day 4, the mice were sacrificed and heart blood was collected into lithium heparin-coated tubes for hematological analysis. The blood was centrifuged and the plasma was separated for biochemical analysis. Subsequently, the liver was removed, weighed and kept at −80 °C for further analysis. The liver weight index was determined by dividing the dry weight of liver by the BW of the subjects.

#### 2.2.6. Hematological Parameter Analysis

Complete blood count, including WBC numbers, percentage of differential WBC, RBC numbers, Hb concentration, Hct value, MCV, MCH and MCHC, were determined using an automatic cell counter (Beckman Coulter Diagnostics, Brea, CA, USA) according to the manufacturer’s instructions at the Medical Technology Laboratory (WU-MeT), Walailak University, Thailand.

#### 2.2.7. Measurement of Lipid-Peroxidation Product

Lipid peroxidation products, including malonyldialdehyde, can react with a TBA reagent and form a TBARS product that is pink in color [29], as has been described by Koonyosying and colleagues. Dry liver tissue samples (10 mg) were manually homogenized in 50 mM PBS pH 2.8 buffer (0.8 mL) BHT (50 ppm in methanol) (0.1 mL). The homogenate (0.1 mL) was combined with 10% (*w*/*v*) TCA (0.22 mL), heated at 90 °C for 30 min, cooled down, centrifuged at 6000 rpm for 10 min and the supernatant was then collected. Plasma was mixed with deproteinizing solution containing 10% TCA and 50 mg/L (*w*/*v*) BHT and then centrifuged at 6000 rpm for 10 min and the supernatant was collected. Supernatants (50 µL) were combined with 0.44 M H_3_PO_4_ (0.15 mL) and 0.6% (*w*/*v*) TBA (0.1 mL), incubated at 90 °C for 30 min and cooled down at 4 °C for 10 min. After that, 1-butanol (0.3 mL) was added to extract the pink-color TBARS product from the reaction solution and it was centrifuged. The supernatant was then collected to photometrically measure optical density (OD) at 540 nm. TBARS concentrations were determined from a calibration curve of standard TMP that had been made by plotting OD_540 nm_ values against TMP concentrations.

#### 2.2.8. Measurement of AST, ALT and ALP Activities

In the test principle, AST catalyzed the reaction from L-aspartate and α-ketoglutarate to oxaloacetate and glutamate. When the produced oxaloacetate was converted into malate by malate dehydrogenase, the nicotinamide adenine dinucleotide was reduced (NADH) and was converted into NAD^+^ (oxidized form) with a decrease in OD of the NADH at 340 nm, which was directly proportional to the AST activity [30]. Similary, ALT catalyzed the reaction from L-alanine and α-ketoglutarate to pyruvate and glutamate. When the produced pyruvate was converted into lactate by lactate dehydrogenase, NADH was converted into NAD^+^ with a decrease in OD of the NADH at 340 nm, which was directly proportional to the ALT activity [30]. In the presence of magnesium and zinc^2+^, ALP hydrolyzed *p*-nitrophenyl phosphate into phosphate and *p*-nitrophenol, which was then determined by measuring the increase in OD at 409 nm. The *p*-nitrophenol that was released was directly proportional to the catalytic ALP activity [31]. In the assay, plasma samples obtained from PbANKA-infected mice and normal mice were diluted with distilled water (1:50) and applied to the Automated ClinChem Analyzer (Roche/Hitachi Cobas C501, Roche Company, Mannheim, Germany) and AST, ALT and ALP activities were assayed using specific assay reagent kits (Roche Company, Mannheim, Germany) according to the manufacturer’s operating instructions. This step was undertaken at the Center for Medical Excellence, Faculty of Medicine, Chiang Mai University.

#### 2.2.9. Statistical Analysis

Data are expressed as mean ± standard error of mean (SEM) values. Statistical significance was analyzed using SPSS program (SPSS version 18.0 for IBM, Chicago, IL, USA licensed by Chiang Mai University, Thailand). Statistical significance was analyzed using one-way analysis of variance test with post hoc Tukey-Kramer test, wherein *p* < 0.05 was considered significantly difference.

## 3. Results

### 3.1. Malaria Suppressive Activity of DFP-RVT in PbANKA-Infected Mice

Suppressive activity levels have been presented as percentages of parasitemia in *P. berghei* (ANKA strain)–infected (PbANKA) mice and relevant suppressive percentage of those mice (Figure 2). In accordance with a placebo treatment, parasitemia in PbANKA mice + dimethyl sulfoxide (DMSO) was found to be 20.45 ± 1.48%. Expectedly, PYR (2 mg/kg body weight (BW)), which is an anti-folate type of anti-malarial drug, clearly exhibited the most effective degree of activity against PbANKA growth (0% parasitemia) in mice (100% suppression) when compared with the placebo group (*p* < 0.05). However, DFP and DFP-RVT exhibited only slight inhibitory effects on PbANKA growth (18.28 ± 1.66 and 15.55 ± 1.13% parasitemia, 18.74 ± 8.12 and 23.96 ± 3.18% suppression, respectively), for which DFP-RVT tentively displayed more effective protective anti-plasmodial effects than DFP at equal concentrations (50 mg/kg BW) and even less than that of PYR.

### 3.2. Changes of Hematological Parameters in PbANKA-Infected Mice

In fact, plasmodium infection in rodents and humans is known to induce RBC hemolysis at the intraerythrocytic stage and during granulocytosis, which can lead to anemia and inflammation, respectively. Herein, levels of white blood cell (WBC) numbers and the percentage of neutrophil were significantly increased in PbANKA mice when compared with normal mice (*p* < 0.05). Notably, PYR exhibited the most potent levels of inhibition against *P. berghei* infection in mice. Accordingly, PYR was found to lower WBC levels and differential neutrophils values, along with increasing differential lymphocytes and monocytes, whereas DFP and DFP-PYR were found to be less effective in this manner (Table 1a). With regard to the RBC indices presented in Table 1b, RBC numbers, hemoglobin (Hb) concentration levels and hematocrit (Hct) levels decreased in PbANKA mice when compared with normal mice (*p* < 0.05). Not surprisingly, PYR was effective at inhibiting *P. berghei* growth in infected mice, while consequently maintaining RBC numbers and Hb concentrations and Hct values at normal levels; whereas, DFP and DFP-RVT were found to be only slightly effective. Moreover, all of the drugs did not influence the mean corpuscular hemoglobin (MCH) values, mean corpuscular hemoglobin concentrations (MCHC), mean corpuscular volume (MCV) and red cell distribution width (RDW).

### 3.3. Inhibitory Effect of Lipid Peroxidation in PbANKA-Infected Mice

Until now, malaria infection has been known to induce ROS production and the oxidation of biomolecules. Accordingly, high increases in lipid-peroxidation products, including thiobarbituric acid-reactive substances (TBARS) (e.g., malondialdehyde) in the plasma (Figure 3a) and livers (Figure 3b) of PbANKA mice (*p* < 0.05), were observed when compared with normal mice. Fantastically, DFP, DFP-RVT and PYR treatments could then restore the increases in TBARS levels in the plasma and livers (*p* < 0.05) of PbANKA mice, for which the orders were PYR > DFP-RVT > DFP in the plasma and DFP-RVT > PYR > DFP in the livers.

### 3.4. Liver Function Enzymes in PbANKA-Infected Mice

Plasma levels of AST, ALT and ALP activities in PbANKA-infected mice were higher than those in normal mice (*p* < 0.05) (Figure 4a–c). However, AST was significantly decreased by DFP treatment when compared to PbANKA mice treated with DMSO, but not by treatment with DFP-RVT nor PYR. Plasma ALT was significantly decreased by PYR treatment when compared to PbANKA mice treated with DMSO, whereas ALP activity was slightly decreased by treatment with DFP, DFP-RVT and PYR when compared to PbANKA mice treated with DMSO.

### 3.5. Body and Liver Weight Changes in PbANKA-Infected Mice

PbANKA infection resulted in a significant decrease in host BW, while subsequent treatments of the PbANKA mice with DFP, DFP-RVT and PYR slightly lowered the host BW (Figure 5a). Conversely, PbANKA infection resulted in a significant increase in the weight of the livers of the hosts, while all the treatments tended to restore increases in the weights of the livers (Figure 5b). Likewise, liver weight index values in PbANKA mice were significantly higher than those in normal mice but were not influenced by any treatments (Figure 5c).

## 4. Discussion

Like others living organisms, malarial parasites require iron. They essentially obtain this iron from their hosts for growth and replication. Conversely, the hosts have to develop defensive mechanisms against the growth of these parasites by withholding iron from them [32]. Thus, the deprivation of iron with the use of an iron chelator is a potential strategy for the treatment of malaria infections. Previous studies have revealed the ability of iron chelators, including DFO, DFP and CM1, to inhibit the growth of malarial parasites [13,14,15,19,28]. In this study, DFP-RVT, as an orally active anti-oxidative iron chelating agent, was tested for its anti-malarial activity in *P. berghei*-infected mice. Interestingly, we found that DFP and DFP-RVT significantly reduced the percentage of parasitemia and resulted in increased rates of survival in parasite-infected mice, for which DFP-RVT exhibited greater potential than DFP. In fact, the lipophilicity and efficacy of anti-malarial drugs are the key factors/determinants for effective malarial treatment. Accordingly, lipophilic low-molecular-weight drugs or compounds would affect their penetrability through the host RBC membrane and the plasmodial membrane, resulting in an effective degree of inhibition for malarial parasitic growth [33,34]. In addition, the ability of drugs to withhold iron from malarial parasites is a critical point for this suggested treatment [35]. We have previously reported that DFP-RVT exhibited greater lipophilicity and higher efficiency in the protection of β1-42 peptide aggregation in lipid-rich neuronal cells and the inhibition of cultured *P. falciparum* growth than DFP [16,23]. In addition, the iron-chelating activity (pFe) of DFP-RVT was found to be stronger than DFP (pFe(III) = 19.6 and 20.6, respectively) [23]. Hence, we confirm that DFP-RVT could possess more potent anti-plasmodial activity against malarial parasitic growth than DFP.

Hepatic enlargement can occur in hosts with malaria infection because the liver is an important target organ during the hepatic stage of the malarial parasite’s life cycle [36,37,38]. In defense of this, parasitized RBC are removed from the blood circulation by phagocytic cells such as Kupffer’s cells in the liver; consequently, liver enlargement has been associated with the severity of malaria infections [39]. Herein, the liver weight indices in all groups of mice were found to correlate with the percentage degree of parasitic growth (expressed in % parasitemia). The weight of the livers obtained from DFP, DFP-RVT and PYR-treated mice were less than those of untreated mice. Apart from the liver weight index values, the hematological parameters were determined to be related to the percentage of parasitic growth, for which WBC numbers could indicate certain underlying conditions such as infection and inflammation [40]. In addition, we found that the iron chelators, DFP and DFP-RVT, maintained WBC numbers within the normal range found in healthy mice. This outcome was similar to that of the PYR treatment group. A previous study has reported that RBC indices were directly influenced by blood-stage malaria infections since the parasitized RBC were rapidly disrupted and cleared from the blood [41]. Consistently, our results have shown that RBC indices, including RBC numbers, Hb concentrations and Hct values, in malaria-infected mice without treatment were significantly decreased and were lower than those of the normal control group. Treatments of PbANKA-infected mice with DFP and DFP-RVT could protect against the destruction of infected RBC. These treatments would also likely prevent RBC from parasitic invasion and eventually improve the anemic conditions caused by malaria parasite-induced RBC hemolysis.

Levels of lipid-peroxidation products, such as MDA and TBARS, reflect the occurrence of oxidative stress that consequently leads to malarial pathogenesis and other related complications. Previous studies have reported that elevated levels of MDA can be recognized as a potential indicator for the severity of malaria infection [22,42]. Alternatively, *P. yoelii* infection was found to reduce the oxidative stress associated with infected RBC and hepatocytes of infected mice, whereas PYR treatment was able to increase their gene expressions and the production of superoxide dismutase and glutathione peroxidase through immune-modulatory mechanisms that could then eliminate the parasites [43]. In the present study, the levels of MDA, which were represented by TBARS concentrations, were determined in the plasma and liver tissues obtained from PbANKA-infected mice that had undergone DFP, DFP-RVT and PYR treatments. These treatments were administered as a way of assessing oxidative stress in the blood and livers of infected mice and to monitor the severity of the malaria infection. Importantly, DFP-RVT possessing one antioxidant part and one iron-chelating site significantly lowered the levels of plasma lipid-peroxidation products to a greater degree than DFP. Extraordinarily, DFP-RVT was determined to be better than DFP and PYR in reducing lipid peroxidation reactions in the livers of infected mice. This was due to a greater degree of lipophilicity, a higher degree of bioavailability in the body, greater accessibility to liver cells and potentially more potent anti-plasmodial activity belonging to the DFP-RVT compound. Taken together, the findings confirm that DFP-RVT could inhibit plasmodium growth in mice, protect against RBC hemolysis and alleviate the inflammation that is associated with parasitic infection, as well as displaying an ability to ameliorate lipid peroxidation in plasma compartments and the liver.

Unlike antioxidant enzymes, plasma levels of AST, ALT and ALP activities are increased during malaria infection. Consequently, an effective anti-malarial treatment can reduce these activities. Agrawal and colleagues have reported that oral PYR treatment (10 mg/kg BW) in *P. yoelii*-infected mice decreased levels of plasma AST, ALT and ALP activities to almost normal levels, whereas levels of glutamine synthetase, glutamate dehydrogenase, monoamine oxidase, carbamoyl phosphates synthetase and ornithine transcarbamoylase were increased [6,7]. Interestingly, dexrazoxane as an iron chelator product inhibited the progression of *P. yoelli* from sporozoites to schizonts in mouse hepatocyte cultures, while methyl-anthranilic DFO reduced the degree of proliferation of *P. falciparum* in HepG2 cells [44,45]. With regard to the elevation of serum ALT activity, liver dysfunctions within a range from mild symptoms to acute hepatitis have been reported in patients with acute *P. falciparum* malaria [46]. Our new findings in animal models have revealed that DFP-RVT, as well as DFP and PYR, slightly decreased plasma AST, ALT and ALP activities in *P. berghei*-infected mice during a 4-day infection period. This would suggest an hepatic anti-inflammatory effect on malaria infection. In addition, hepatic mitochondrial cytochrome P-450 and cytochrome b5 activities were significantly depressed in *P. yoelii*-infected mice and restored to almost normal levels by oral PYR treatments (10 mg/kg BW) [47].

## 5. Conclusions

The lipophilic DFP-RVT hybrid exerted more potent anti-plasmodial and preventive effects on *P. berghei* infection in mice than DFP. This compound was found to improve anemia through decreased hemolysis, inhibit lipid peroxidations and alleviate inflammation in the livers of mice diagnosed with malaria infections. Taken together, DFP-RVT would be a promising orally active iron-chelating agent that exhibits anti-plasmodial, anti-hemolysis and anti-lipid peroxidation effects on malarial parasitic infection. Consequently, effective and safe doses of the DFP-RVT compounds are of significant interest for further investigations involving animals and human subjects.

## Figures and Tables

**Figure 1 biology-10-00911-f001:**
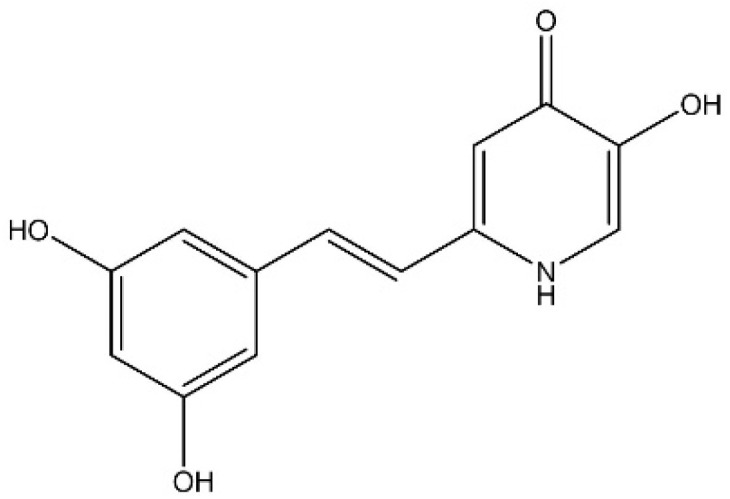
Chemical structure of DFP-RVT hybrid (Redrawn from [23]).

**Figure 2 biology-10-00911-f002:**
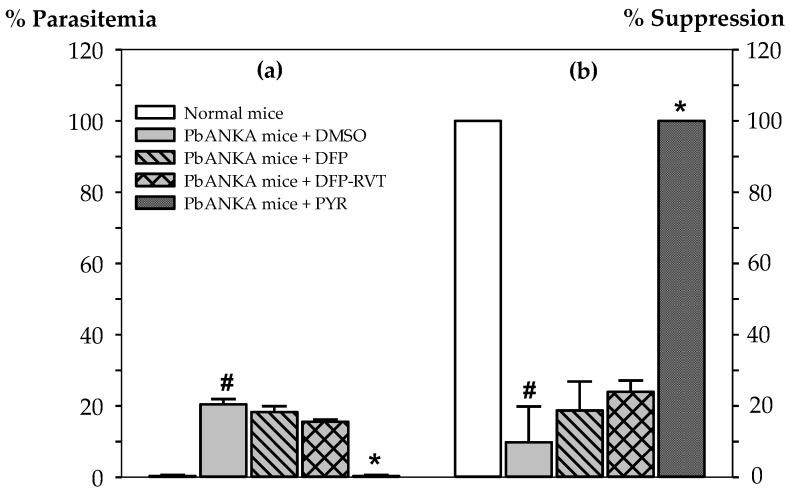
Suppressive activity of DFP, DFP-RVT and PYR against *P. berghei* (ANKA strain) infection in mice. Male ICR mice (n = 3 each) were infected intraperitoneally with PRBC suspension (1 × 107 cells/mL with 5–20% parasitemia), then orally administered with 60% DMSO, PYR (2 mg/kg BW), DFP (50 mg/kg BW) and DFP-RVT (50 mg/kg BW) every day. The BW of the subjects were recorded after 4 d and they were finally euthanized. Blood was collected from the hearts of the subjects and the percentage of PRBC was determined and expressed as either % parasitemia (**a**) or % suppression (**b**). Data are expressed as mean ± SEM values (n = 3). ^#^
*p* < 0.05 when compared to normal mice; * *p* < 0.05 when compared to PbANKA mice treated with DMSO.

**Figure 3 biology-10-00911-f003:**
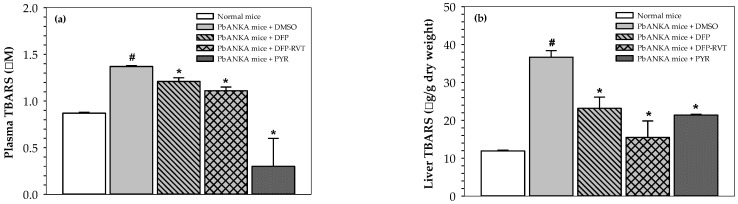
Levels of TBARS in plasma and liver tissues collected from normal mice and PbANKA-infected mice treated with DMSO, DFP, DFP-RVT and PYR. Male ICRmice were infected *ip* with PRBC suspension (1 × 10^7^ cells/mL with 5–20% parasitemia), then orally administered with 60% DMSO, PYR (2 mg/kg BW), DFP (50 mg/kg BW) and DFP-RVT (50 mg/kg BW) every day for 4 d. The mice were then humanely euthanized. Heart blood was collected into lithium heparin-coated tubes and plasma was separated for determination of TBARS concentration (**a**). Livers were removed, homogenized and TBARS concentrations were determined (**b**). Data are expressed as mean ± SEM values. ^#^
*p* < 0.05 when compared to normal mice; * *p* < 0.05 when compared to PbANKA mice treated with DMSO.

**Figure 4 biology-10-00911-f004:**
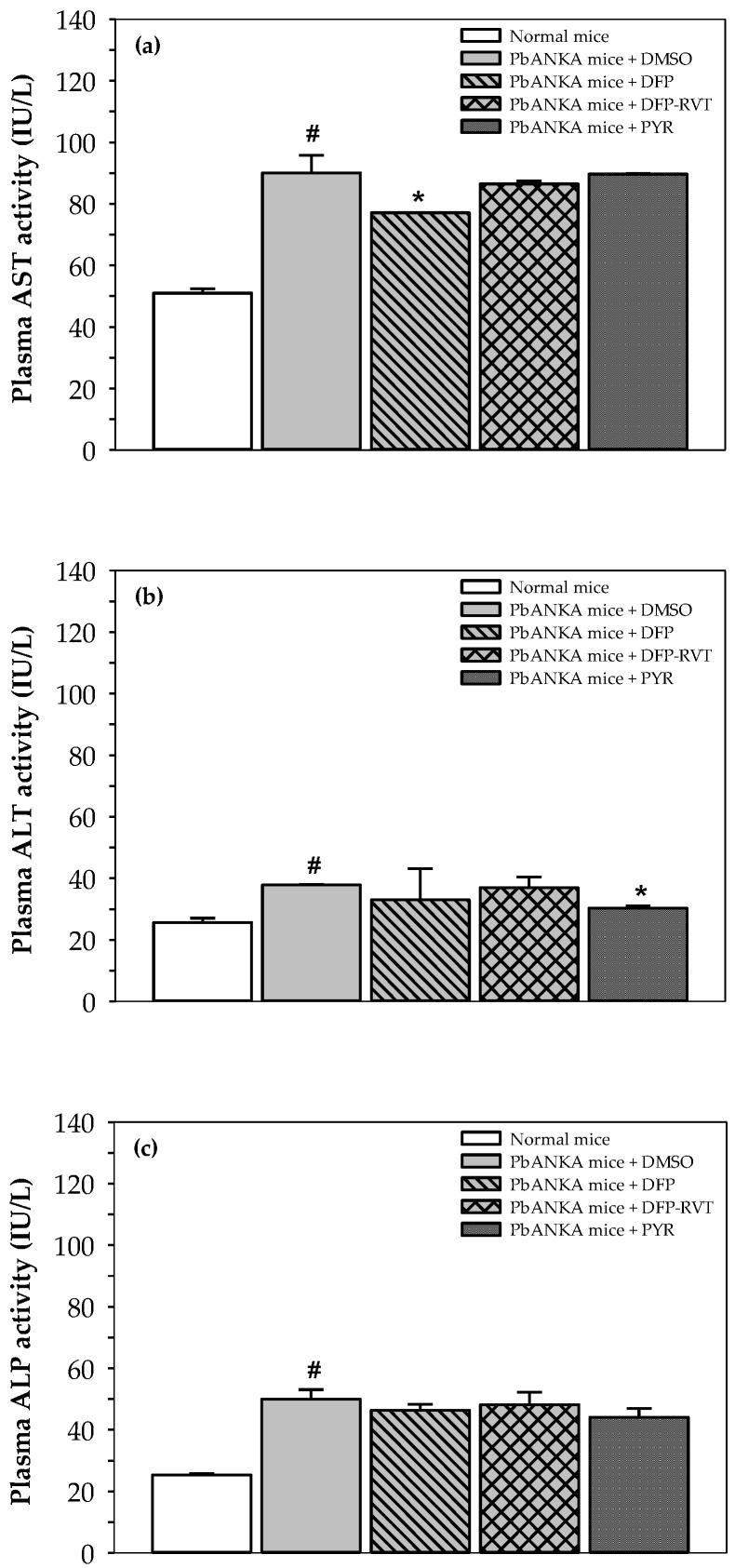
Levels of AST, ALT and ALP activities in plasma collected from normal mice and PbANKA-infected mice treated with DMSO, DFP, DFP-RVT and PYR. (**a**) Plasma AST activity, (**b**) Plasma ALT activity, and (**c**) Plasma ALP activity. Male ICR mice were infected *ip* with PRBC suspension (1 × 10^7^ cells/mL with 5–20% parasitemia), then orally administered with 60% DMSO, PYR (2 mg/kg BW), DFP (50 mg/kg BW) and DFP-RVT (50 mg/kg BW) every day for 4 d. The mice were then humanely euthanized. Heart blood was collected into lithium heparin-coated tubes and plasma was separated for analysis of aspartate aminotransferase (AST), alanine aminotransferase (ALT) and alkaline phosphatase (ALP) activities. Data are expressed as mean ± SEM values. ^#^
*p* < 0.05 when compared to normal mice; * *p* < 0.05 when compared to PbANKA mice treated with DMSO.

**Figure 5 biology-10-00911-f005:**
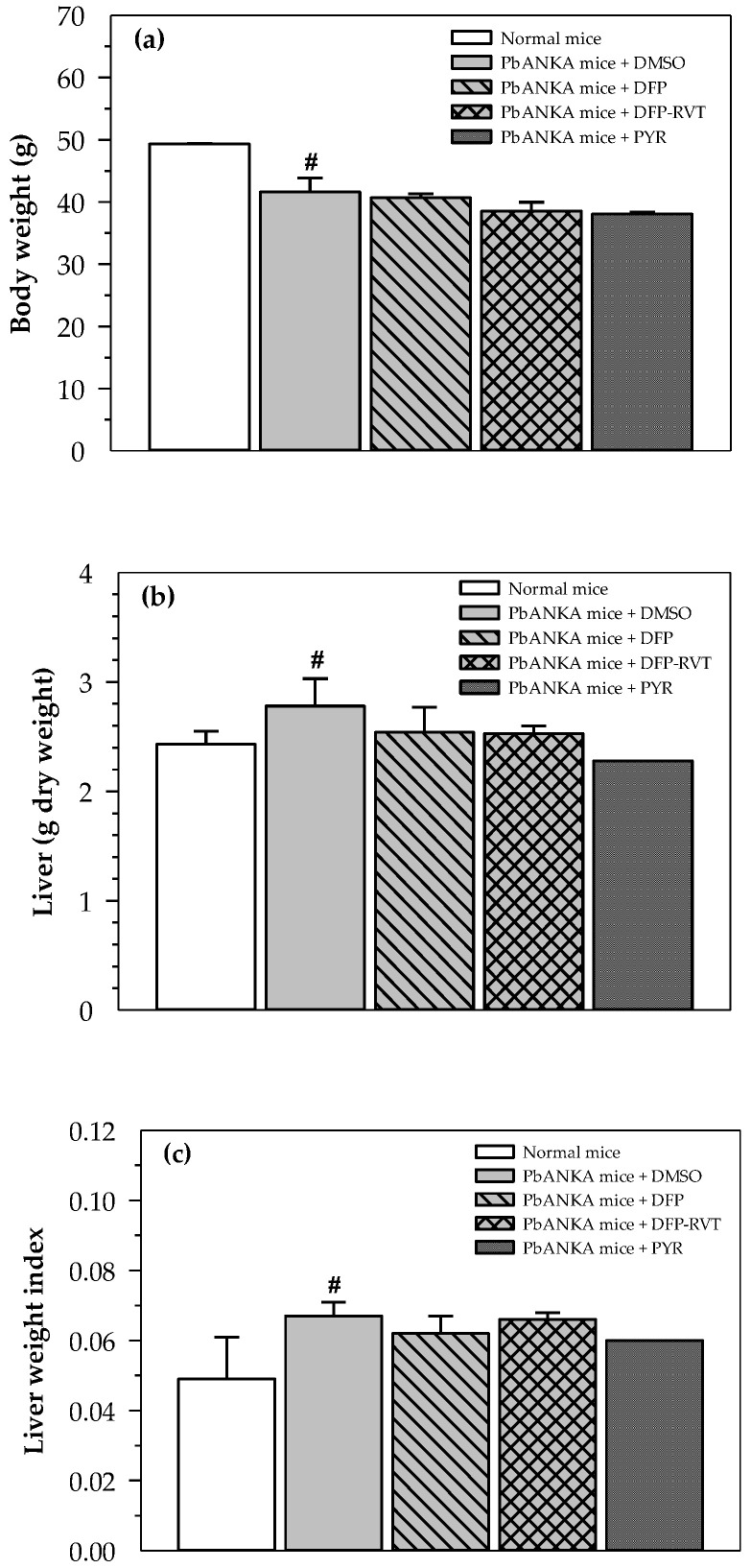
Body and liver weight, and liver weight index of normal mice and PbANKA-infected mice treated with DMSO, DFP, DFP-RVT and PYR. Male ICR mice were infected *ip* with PRBC suspension (1 × 10^7^ cells/mL with 5–20% parasitemia), then orally administered with 60% DMSO, PYR (2 mg/kg BW), DFP (50 mg/kg BW) and DFP-RVT (50 mg/kg BW) every day for 4 d. Mice were then humanely euthanized. Body weights were recorded (**a**), while livers were removed and weighed (**b**), and converted to liver weight index (**c**). Data are expressed as mean ± SEM values. ^#^
*p* < 0.05 when compared to normal mice.

**Table 1 biology-10-00911-t001:** Levels of hematological parameters in blood obtained from normal mice and PbANKA-infected mice treated with DMSO, DFP, DFP-RVT and PYR. Male ICR mice (n = 3 each) were infected *ip* with PRBC suspension (1 × 10^7^ cells/mL with 5–20% parasitemia), then orally administered with 60% DMSO, PYR (2 mg/kg BW), DFP (50 mg/kg BW) and DFP-RVT (50 mg/kg BW) every day for 4 d. The mice were then humanely euthanized. Heart blood was collected into lithium heparin-coated tubes and were analyzed in terms of total and differential WBC numbers and RBC indices. Data are expressed as mean ± SEM values (n = 3). ^#^
*p* < 0.05 when compared to normal mice; * *p* < 0.05 when compared to PbANKA mice treated with DMSO.

**(a) Total and Differential WBC Coun**
**Mice/Treatment**	**WBC** **Numbers** **(×10^3^/mL)**	**Differential WBC (%)**	**Mice/Treatment**	**WBC Numbers** **(×10^3^/mL)**	**Differential** **WBC (%)**	**Mice/Treatment**	**-**
		**Neutrophil**			**Neutrophil**		**-**
Normal/DI	3.38 ± 0.33	22.0 ± 0.6	74.0 ± 0.4	4.0 ± 0.0	0	0	-
PbANKA/60% DMSO	10.18 ± 0.04 ^#^	70.0 ± 6.5 ^#^	33.7 ± 8.7	3.0 ± 0.0	1.0 ± 0.6	0.0 ± 0.0	-
PbANKA/DFP (50 mg/kg BW)	3.25 ± 0.07 *	17.0 ± 2.4 *	37.7 ± 9.9	25.7 ± 12.2	1.0 ± 0.6	0.3 ± 0.3	-
PbANKA/DFP-RVT (50 mg/kg BW)	4.38 ± 0.44 *	64.5 ± 4.5	27.0 ± 5.3	2.7 ± 0.3	1.0 ± 0.0	0.3 ± 0.3	-
PbANKA/PYR (2 mg/kg BW)	2.81 ± 0.07 *	29.0 ± 1.6 *	57.5 ± 2.9 *	12.0 ± 4.9*	1.0 ± 0.0	0.5 ± 0.4	-
**(b) RBC Indices**
**Mice/Treatment**	**RBC** **Numbers** **(×10^6^/mL)**	**Hb (g/dL)**	**Hct (%)**	**MCV (fL)**	**MCH (pg)**	**MCHC (g/dL)**	**RDW (%)**
Normal/DI	9.70 ± 0.01	14.70 ± 0.3	51.8 ± 0.4	53.0 ± 0.0	15.0 ± 0.0	28.0 ± 0.0	15.0 ± 0.0
PbANKA/60% DMSO	7.03 ± 0.07 ^#^	12.6 ± 0.0 ^#^	44.0 ± 0.0 ^#^	62.7 ± 0.7 ^#^	18.0 ± 0.0 ^#^	28.7 ± 0.3	15.0 ± 0.6
PbANKA/DFP (50 mg/kg BW)	7.72 ± 0.49 *	13.2 ± 3.9	46.0 ± 2.4	60.3 ± 0.7	17.7 ± 0.7	29.3 ± 0.0	14.0 ± 0.6
PbANKA/DFP-RVT (50 mg/kg BW)	7.91 ± 0.41 *	13.6 ± 0.5	47.5 ± 0.4	60.7 ± 2.6	17.3 ± 0.3	28.7 ± 1.2	14.7 ± 0.3
PbANKA/PYR (2 mg/kg BW)	9.19 ± 0.91 *	14.6 ± 1.6 *	49.5 ± 5.3	61.0 ± 0.0	18.0 ± 0.0	30.0 ± 0.0	14.5 ± 1.2

BW = body weight, DI = deionized water, DMSO = dimethylsulfoxide, DFP = deferiprone, DFP-RVT = deferiprone-resveratrol hybrid, Hb = hemoglobin, Hct = hematocrit, MCH = mean corpuscular hemoglobin, MCHC = mean corpuscular hemoglobin concentration, MCV = mean corpuscular volume, PbANKA = *P. berghei* (ANKA strain)-infected mice, PYR = pyrimethamine, RBC = red blood cells, RDW = red cell distribution width, WBC = white blood cells.

## Data Availability

The authors confirm that the data supporting the findings of this study are available within the article.

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
