# Peer review of "Anti-Malarial and Anti-Lipid Peroxidation Activities of Deferiprone-Resveratrol Hybrid in Plasmodium berghei-Infected Mice"

_biology, 2021, doi:10.3390/biology10090911_

Round 1

Reviewer 1 Report

The topic is worthy of publication in light of the increasing drug resistance to malaria. The paper was properly organized and the theses posed were presented and reflected in the results. The inhibitory effect of the DFP-RVT hybrid on parasite growth and oxidative stress levels in P. berghei infected mice was demonstrated. Based on the mouse study, it can be concluded that lipophilicity and efficacy of antimalarial drugs are key factors/determinants for successful malaria treatment.
The material and methods are informative and can be replicated by others. 
The presentation of the results is very good. 
The literature is adequate for the subject.

There are few editorial errors, which I have highlighted in yellow in the text

Reviewer 2 Report

The manuscript entitled “Anti-Malarial and Anti-Lipid Peroxidation Activities of Deferiprone-Resveratrol Hybrid in Plasmodium berghei-Infected Mice” is about to evaluate the effects of Deferiprone-Resveratrol Hybrid on Malaria.

The manuscript is need some modification to be published in this journal.

Authors need clarify the “The percentages of suppression of parasitic growth or %survival” at line 154, because I think the concept of percentages of suppression of parasitic growth is not same with %survival. And also authors need consistently use same thing like percentage or % in entire manuscript.

Line 183, form should be from.

Line 191, ALT should be AST.

Line 198, Authors need clarify what is the plasma.

Line 212: I recommend that some of subtitle should be changed because some of them did not showed what does that mean, for example, “Suppressive Activity of DFP-RVT in PbANKA-Infected Mice”, what is suppressed by DFP-RVT?

Line 286, Authors need to use consistently for everything in manuscript. For example, in “Liver function enzymes in PbANKA-Infected Mice” function and enzyme should be capitalized because authors use for capital letter in other subtitle.

I found several times that the sentence were written not clearly. For example, “the corresponding increase in AST was significantly decreased by DFP treatment, but not by treatment with DFP-RVT nor PYR, while the corresponding increase in ALT was significantly decreased by PYR treatment.” I understand what authors want to say but, it is not clear what does “the corresponding increase” mean. So authors need to carefully check the manuscript to remove that kind of problem.

Finally, I found some unconsistently written reference, So authors need double check that.

Reviewer 3 Report

The authors report a study aimed at showing the antimalarial properties of a deferiprone-resveratrol hybrid, which could be used as an adjuvant treatment against malaria.

  1. List of abbrevations : Some abbreviations in the list at the end of the manuscript are not used in the text, whereas others, such as "LIP","TMB", "MDA" or "ALP" are used and are not defined. So please modify the list of abbreviations to ensure that it corresponds precisely to the abbreviations used in the text.
  2. The structure of the deferiprone-resveratrol hybrid used in this study is not clear : (i) the structures shown in the graphical abstract and in the Figure 1 are not identical, and do not correspond to the name given on line 92; (ii) according to the original publication by Xu et al. (2017) (i.e. reference 23), the best deferiprone-resveratrol hybrid would be 2-(4-ethoxystyryl)-5-hydroxy-1-methyl-4(1H)-pyridinone. In particular the structure given in Figure 1 would not be quite lipophilic, a property that should be important for its antimalarial activity, as indicated in this manuscript. So please clarify.
  3. Introduction, line 79 : Actually ROS scavengers, and not ROS, would be beneficial for the host by defending against parasitic growth without causing oxidative cell damage.
  4. What are IRC mice ?
  5. Methods (lines 183-185) : The abbreviation "TMB" is not defined, so it is not possible to understand how TBARS concentrations were determined. For instance if a calibration curve using malondialdehyde (MDA) was performed, so TBARS may be indicated as MDA equivalent. So please clarify to what TBARS concentrations correspond.
  6. Discussion, line 401 : Attention, plasma AST, ALT and ALP activities are increased during malaria, unlike antioxidant enzymes, so an effective antimalarial treatment reduces these activities. Please clarify.
  7. English should be improved and checked for typing mistakes.
